# Control of Oxidative Stress in Cancer Chemoresistance: Spotlight on Nrf2 Role

**DOI:** 10.3390/antiox10040510

**Published:** 2021-03-25

**Authors:** Giuseppina Barrera, Marie Angele Cucci, Margherita Grattarola, Chiara Dianzani, Giuliana Muzio, Stefania Pizzimenti

**Affiliations:** 1Department of Clinical and Biological Sciences, University of Turin, Corso Raffaello 30, 10125 Turin, Italy; marieangele.cucci@unito.it (M.A.C.); margherita.grattarola@unito.it (M.G.); giuliana.muzio@unito.it (G.M.); stefania.pizzimenti@unito.it (S.P.); 2Department of Scienza e Tecnologia del Farmaco, University of Turin, Via Pietro Giuria 11, 10125 Turin, Italy; chiara.dianzani@unito.it

**Keywords:** oxidative stress, chemoresistance, (NF-E2-related factor 2) Nrf2, antioxidant transcription factors

## Abstract

Chemoresistance represents the main obstacle to cancer treatment with both conventional and targeted therapy. Beyond specific molecular alterations, which can lead to targeted therapy, metabolic remodeling, including the control of redox status, plays an important role in cancer cell survival following therapy. Although cancer cells generally have a high basal reactive oxygen species (ROS) level, which makes them more susceptible than normal cells to a further increase of ROS, chemoresistant cancer cells become highly adapted to intrinsic or drug-induced oxidative stress by upregulating their antioxidant systems. The antioxidant response is principally mediated by the transcription factor Nrf2, which has been considered the master regulator of antioxidant and cytoprotective genes. Nrf2 expression is often increased in several types of chemoresistant cancer cells, and its expression is mediated by diverse mechanisms. In addition to Nrf2, other transcription factors and transcriptional coactivators can participate to maintain the high antioxidant levels in chemo and radio-resistant cancer cells. The control of expression and function of these molecules has been recently deepened to identify which of these could be used as a new therapeutic target in the treatment of tumors resistant to conventional therapy. In this review, we report the more recent advances in the study of Nrf2 regulation in chemoresistant cancers and the role played by other transcription factors and transcriptional coactivators in the control of antioxidant responses in chemoresistant cancer cells.

## 1. Introduction

Drug resistance, which occurs in nearly all types of cancer, is a major problem in the treatment of cancer patients. Drug resistance can be classified in two ways—the intrinsic resistance, when tumors are resistant prior to treatment, and therefore, the drugs are not effective, even with initial early diagnosis and treatment, and acquired resistance, which occurs after prolonged cycles of chemotherapy, despite an initial positive response [1]. The intrinsic resistance includes aberrations of signals downstream or upstream of the targeted proteins leading to the acquisition of cancer hallmarks, while acquired resistance is inclined to preserve the original alterations and develop additional alterations that can adapt cancer cells to resist further drug treatments [2]. Beyond specific molecular alterations, which can lead to targeted therapy, a mountain of evidence has suggested that the regulation of redox status plays an important role in cancer cell survival to the therapy [3,4]. In recent years, the interest in the role played by oxidative stress in cancer and chemoresistance has increased enormously, as shown by the growing number of scientific publications devoted to these aspects (Figure 1).

In this review, we examined diverse molecular and biochemical changes involved in increasing the antioxidant potential of chemoresistant tumors.

## 2. Oxidative Stress in Cancer Cells

Oxidative stress is involved in various and numerous pathological states including cancer, chronic inflammation, atherosclerosis, and neurodegenerative diseases. Oxidative stress describes the imbalances in redox couples, such as reduced to oxidized glutathione (GSH/GSSG) or NADPH/NADP+ ratios, leading to an increase of molecules enriched with one or more oxygen atoms that are considered markers of oxidative stress [5]. The most important representatives of these reactive oxygen species (ROS) are superoxide anion (O_2_^•−^), perhydroxyl radical (HO_2_^•^), hydrogen peroxide (H_2_O_2_), and hydroxyl radical (^•^OH) [6]. ROS can be generated by either enzymatic or non-enzymatic processes. Enzymatic ROS formation includes the production of superoxide anion from the mitochondrial electron transport chain, or its production via nicotinamide adenine dinucleotide phosphate (NADPH) oxidases, xanthine oxidase, and cyclooxygenases. Non-enzymatic formation of ROS comprises the generation of hydroxyl radicals from H_2_O_2_ in the Fenton or Haber–Weiss reactions [7]. To maintain the ROS at a nontoxic level, cells possess both small molecules, such as glutathione and urate, and several antioxidant enzymes such as superoxide dismutases, MnSOD, and Cu/ZnSOD, which are located in the mitochondria and the cytosol, respectively [8], catalase, glutathione peroxidase (GPX), glutathione reductase, and heme oxygenase (HO) [6]. ROS are a major cause of the alteration of macromolecules and can influence various cancer hallmarks acting as a double-edged sword in cancer [9]. Indeed, a small increase in oxidative stress is functional for cancer cells since ROS are essential for various biological functions, including cell survival and proliferation. Generally, cancer cells present a ROS level higher than normal cells due to ROS of non-mitochondrial origin [10], pro-growth and pro-inflammatory factors [11], dysfunction of mitochondria or peroxisomes [12], altered metabolism, and hypoxia [13]. At the same time, cancer cells have a marked antioxidant capacity that helps them in maintaining the redox balance at nontoxic levels (Figure 2). On the other hand, the high basal ROS levels make cancer cells more susceptible than normal cells to a further increase of ROS [14]. Indeed, a ROS increase may push cancer cells beyond the life threshold, leading to the activation of different cell death pathways, thus limiting the cancer progression [15]. Among these pathways, a mountain of evidence strongly suggests that ROS-induced autophagic cell death may play an important role [16,17]. Indeed, it has been demonstrated that ROS upregulates the expression of autophagy-related genes (ATGs), such as beclin-1 [18] and ATG4 [19].

On the basis of these observations, the use of radiation therapy or drugs that further induce ROS in cancer cells has been explored as a therapeutic option [20]. Many chemotherapeutic drugs, such as cisplatin, etoposide, paclitaxel, doxorubicin, and bortezomib, in addition to other mechanisms, kill cancer cells by inducing high ROS levels. Other drugs, able to deplete reduced glutathione (GSH) or inhibit antioxidant enzymes, such as superoxide dismutase (SOD) or thioredoxin reductase (TRX), have been tested as anticancer agents [21]. Recently, anticancer therapies that induce oxidative stress by increasing the amount of ROS and/or inhibiting antioxidant levels have received great attention.

## 3. Oxidative Stress and Chemoresistance

Chemoresistant cancer cells become highly adapted to intrinsic or drug-induced oxitive stress by upregulating their antioxidant systems [22,23] (Figure 2). In a recent review, Sferrazzo et al. reported that HO-1 overexpression is involved in chemotherapy and radiotherapy resistance of brain tumors such as neuroblastoma, medulloblastoma, meningioma, and astrocytoma [24]. Other studies, regarding non-small-cell lung carcinoma (NSCLC) and small cell lung carcinoma (SCLC) cell lines, demonstrated that high levels of antioxidants were associated with decreased platinum–DNA binding and intracellular platinum accumulation, increasing cisplatin resistance [25,26]. Accordingly, the reduction of cellular antioxidants, namely, GSH content, sensitized cancer cells to cisplatin [27]. The malignant mesothelioma cells also presented higher levels of SOD mRNA and activities, together with elevated catalase and GSH levels, compared to nonmalignant mesothelial cells, and were more resistant to H_2_O_2_ and epirubicin treatments [28]. Analogously, high levels of GSH preserved cells from platinum drug toxicity [29].

Increased antioxidant content was also demonstrated in multidrug-resistant breast cancer cells that presented increased thiols and reduced lipoperoxidation [30]. In agreement with these observations, chemotherapy response in patients with primary breast cancer demonstrated that reduced oxidative stress levels might compromise the effectiveness of adjuvant chemotherapy [31]. The upregulation of xCT, a downstream target gene of Nrf2, responsible for the import of cysteine to support GSH synthesis, is a general feature of chemoresistant breast cancer cells. We demonstrated that the downregulation of xCT, altered cancer stem cells (CSC) intracellular redox balance, suggesting that xCT plays a functional role in CSC biology [32]. In addition, xCT has been implicated in multidrug resistance of lung cancer [33], and in cisplatin resistance in ovarian cancer cells [34].

Although oxidative stress can be one of the major causes underlying hepatocellular carcinoma (HCC) development by driving DNA damage production and altered protein expression, in HCC patients, the increased expression of Nrf2, the master regulator of antioxidant genes, accompanied the increase of 8-hydroxyguanosine (8-OHdG) lesions, which was found to be associated with poor survival [35]. Moreover, in Hepa-1 and HepG2 liver cancer cells, Nrf2 mediated the expression of the antiapoptotic gene Bcl-xL preventing apoptosis and promoting drug resistance [36].

Pancreatic cancer is an extremely aggressive, chemo-resistant tumor, and its invasive and refractory nature to therapeutic intervention makes it one of the worst prognosis cancers. It has been reported that the expression of Nrf2 is upregulated in pancreatic cancer cell lines and ductal adenocarcinomas, indicating a greater intrinsic capacity of these cells to respond to stress signals and to resist chemotherapeutic interventions [37]. Accordingly, the inhibition of Nrf2 signaling, by using digoxin, reversed chemoresistance of gemcitabine-resistant pancreatic cancer [38].

The increase of antioxidant response by Nrf2 is also involved in chemoresistance of colon cancer cells, where the inhibition of either Nrf2 or Her2, alone and in combination, caused a significant increase in oxaliplatin-induced cytotoxicity and apoptosis with a maximum effect in SW480/Res cells having low Her2 and Nrf2 expression levels [39].

In prostate cancer cells, the induction of GSH levels by USP2a, a deubiquitinating enzyme overexpressed in prostate adenocarcinomas, leads to chemoresistance to typically pro-oxidant agents, such as cisplatin, doxorubicin, and docetaxel taxane [40].

In bladder cancer cells, Nrf2 overexpression is associated with clinically relevant cisplatin resistance [41]. In our recent study in bladder cancer cells, we demonstrated that the increase of oxidative stress, linked to the inhibition of Nrf2 expression by treatment with Ailanthone (Aila), a natural active compound isolated from the Ailanthus altissima, inhibited cell growth and migration [42]. Similar results were obtained in cis-diamminedichloroplatinum (CDDP)-resistant ovarian cancer cells, together with the demonstration that Aila acts by increasing Nrf2 protein degradation [43]. Moreover, the use of carbosilane dendrimers loaded with siRNA targeting Nrf2 was able to overcome cisplatin chemoresistance in bladder cancer cells [44].

In melanoma cells, high levels of Nrf2 and GSH have been individuated as key mediators of resistance to temozolomide [45]. An increase of oxidative stress can also reverse the resistance to PLX4032, a specific BRaf inhibitor, in BRaf V600E mutated melanoma cells [46], and block the metastasis of sarcoma [47]. As far as it regards hematological tumors, it has been demonstrated that Nrf2 overexpression in acute myeloid leukemia (AML) cells was able to cause resistance toward cytarabine and daunorubicin, two common anticancer drugs used in this type of leukemia [48].

## 4. Control of Nrf2 Transcription Factor in Chemoresistance

In normal and cancer cells, the antioxidant response is principally mediated by the transcription factor Nrf2 (NF-E2-related factor 2), which has been considered the master regulator of antioxidant and cytoprotective genes [49]. Under physiological conditions, Nrf2 in the cytoplasm is bound by Kelch-like ECH-associated protein 1 (Keap1). Keap1 forms a complex with Cul3 and Rbx1, and this E3 ubiquitin ligase complex binds and ubiquitinates Nrf2, driving Nrf2 to proteasomal degradation [50]. When oxidative stress increases in the cell, the cysteine residues of Keap1 become oxidized, resulting in a conformational change of the Keap1–Nrf2 complex, which prevents Nrf2 ubiquitination [51]. The stabilized Nrf2 translocates in nuclei, heterodimerizes with small Maf proteins, and binds the antioxidant response element (ARE)/electrophile response element (EpRE), located within the promoter region of specific target genes, inducing the expression of a large number of cytoprotective proteins with antioxidant and detoxifying roles [52].

Several studies have shown that the activation of Nrf2 promotes cancer progression [53], invasion, metastasis [54], and chemoresistance [55]. In recent years, the Keap1/Nrf2 system has attracted much interest from scientists in basic and clinical cancer research, which suggested that this system could represent a target for new therapeutic options [49].

### 4.1. Nrf2 and Keap1 Genetic Alterations

As discussed in the above paragraph, several types of chemoresistant tumors display high levels of Nrf2 expression, which often correlates with a worse prognosis. This increase may depend on the dysregulation of the Keap1/Nrf2 system due to the mutation of Keap1 that makes it unable to bind to Nrf2. These mutations have been found in lung [56], gallbladder [57], and ovarian cancers [58]. Mutations in the Nrf2 gene were also found in patients with squamous NSCLC and head and neck carcinoma, which expressed abnormal transcript variants from the NFE2L2 gene (encoding Nrf2), and encoded Nrf2 protein isoforms missing the Keap1 interaction domain. This loss of interaction with Keap1 resulted in an Nrf2 stabilization and induction, and an increase of Nrf2 transcriptional response [59]. Using a variety of computational methods, NFE2L2 mutations have been considered likely to drive lung, esophageal, cervical, bladder, and uterine cancers [60,61]. By analyzing the patterns of mutational signatures in tumors of more than 8200 tumor-normal pairs and by integrating the driver predictions with information on somatic copy number alterations, Davoli et al. have designed Nrf2 as an oncogene [62].

### 4.2. Altered Nrf2 Transcription

Beyond genetic mutations in Keap1 and Nrf2 codifying genes, the high expression of Nrf2 can also be due to the upregulation of NFE2L2 gene transcription by K-Ras^G12D^, B-Raf^V619E,^ and Myc^ERT2^. However, the precise regulatory mechanisms of this increase of Nrf2 transcription are still unknown [53]. It has been reported that Nrf2 can be activated by the K-Ras oncogene, conferring chemoresistance in NSCLC cells [63]. More recently, Liang et al. demonstrated that in K-Ras-driven pancreatic ductal adenocarcinoma (PDAC), Nrf2 expression can be regulated by protein interacting with never in mitosis A1 (PIN), which increases the promoter activity of Nrf2 upregulating its transcription [64]. Another transcription factor involved in Nrf2 expression is Forkhead box M (FOXM1) [65]. Interestingly, overexpression of FOXM1 is a common feature of most human cancers, and it is associated with disease progression and adverse prognosis in multiple human cancer types [66]. Moreover, abnormal activation of FOXM1 also contributes to drug resistance in cancers, including ovarian cancer, breast cancer, prostate cancer, nasopharyngeal carcinoma, acute myeloid leukemia, and colorectal cancer [67,68,69,70,71,72]. Although in these cancers a direct correlation between FOXM1 overexpression and the increase of Nrf2 protein was not investigated, we can postulate that this pathway could also contribute to the overexpression of Nrf2 in such chemoresistant cancers.

A negative interaction between NF-kB transcription factor and Nrf2 promoter has been found by Rushworth et al. [73]. These authors demonstrated that NF-kB suppressed Nrf2 transcription and sensitized myeloid leukemia cells to cytarabine and daunorubicin.

### 4.3. Control of Nrf2 Protein Degradation

Another mechanism involved in Nrf2 overexpression in chemoresistant cancers is the rate of degradation, depending on the balancing between ubiquitination and deubiquitination of Keap1 and Nrf2 proteins. E3 ubiquitin ligase complexes control the ubiquitination and proteasomal degradation of Nrf2 [74]. Recent evidence suggests that alteration in the Cul3–Keap1–E3 ligase complex, due to mutation of Nrf2 or Keap1, leads to constitutive activation of Nrf2, which contributes to chemoresistance [75]. Zhang et al., by screening a deubiquitinase (DUB) library, identified DUB3 as a stabilizer of Nrf2. DUB3 promotes Nrf2 stability and transcriptional activity by decreasing the K48-linked ubiquitination of Nrf2. Moreover, these authors demonstrated that ectopic expression of DUB3 caused Nrf2-dependent chemotherapy resistance in colon cancer cell lines [76]. Conversely, a deubiquitinase, USP15, has been found to deubiquitinate Keap1 and indirectly aggravate Nrf2 Lys-48-linked polyubiquitination, thus decreasing Nrf2 protein level [77].

### 4.4. Epigenetic Regulation of Nrf2 Expression

Extensive research has demonstrated that a potent mechanism of gene expression control is mediated through the function of short noncoding RNAs, especially for microRNAs (miRNAs). Shi et al. reported that Nrf2 was the direct target gene of miR-340, which suppresses the Nrf2-dependent antioxidant pathway, suggesting that lower expression of miR-340 is involved in the development of CDDP resistance in hepatocellular carcinoma [78]. In accordance with these results, it has been demonstrated that miR-141 plays a key role in 5-FU resistance in hepatocellular carcinoma cells by downregulating Keap1 expression, thereby reactivating the Nrf2-dependent antioxidant pathway [79]. Other authors reported that miR-432-3p also downregulated Keap1, thus inducing an increase in Nrf2 activity, and overexpression of miR-432-3p resulted in a decreased sensitivity of esophageal squamous cell carcinoma cells to CDDP [80].

### 4.5. Control of Nrf2 Activity by Bach1

Like Nrf2, the transcription factor for BR-C, ttk and bab (BTB) and cap’n’collar (CNC) homology 1 (Bach1) belonging to the cap’n’collar (CNC) b-Zip family of proteins, binds to AREs in response to changes in redox states [51]. Bach1 functions as a molecular sensor of intracellular heme, tuning transcription to the fluctuation of heme levels [81,82], and, together with Nrf2 and the Maf transcription factors, control the expression of HO-1 and other antioxidant genes [83]. Bach1 competes with Nrf2 for binding to the AREs in oxidative stress-response genes. An increase of oxidative stress leads to the oxidation of heme-containing proteins, which release free heme with consequent degradation of Bach1, which exerts an antioxidant effect [84]. Bach1 may also activate transcription of several genes, including MMP1 and CXCR4, which are involved in inducing breast cancer metastasis [85], and promotes lung cancer metastasis by activating hexokinase 2 and glyceraldehyde 3-phosphate dehydrogenase transcription, and by increasing glucose uptake, glycolysis rates, and lactate secretion [86]. Interestingly, cooperation between Nrf2 and Bach1 has been proposed in lung cancer metastatization. In this cancer model, Nrf2 activated a metastatic program by inhibiting the heme-degradation of Bach1 [54].

The main mechanisms involved in the regulation of Nrf2 expression are illustrated in Figure 3.

## 5. Other Transcription Factors Involved in Oxidative Stress Control

Beyond Nrf2, other players participate to increase the antioxidant potential in chemoresistant cancer cells. Although Nrf2 controls the transcription of several antioxidant genes, it does not regulate directly the classic antioxidant genes SOD1 or SOD2, which instead are controlled by other transcription factors [87].

### 5.1. NF-kB Transcription Factor

A majority of the transcriptional activation of SOD2 is mediated through NF-κB transcription factor [88]. Other authors reported that the antioxidant targets induced by NF-κB include glutathione S-transferase, metallothionein-3, NAD(P)H dehydrogenase [quinone]1, HO-1, and glutathione peroxidase-1 [89]. NF-κB is also involved in the control of inflammation [90], apoptosis [91], epithelial to mesenchymal transition [92], and induction of MDR1/P-gp expression [93]. NF-κB may play a protective role under conditions of oxidative stress by suppressing ROS accumulation and its activation prevents oxidative stress-induced apoptosis by inducing MnSOD and thioredoxin levels [94]. Much evidence over the last few years have indicated that most chemotherapeutic agents activate NF-κB in in vitro and in vivo experiments, and this transcription factor has been identified as a key player in resistance mechanisms [95]. Indeed, NF-kB inhibition increases the sensitivity of cancer cells to the apoptotic action of chemotherapeutic agents [96]. Recent studies showed that NF-κB inhibitors in combination with cytotoxic compounds improved chemotherapy sensitivity in pancreatic cancer [97], and the inhibition of TRAF6/NF-κB p65/P-gp axis by miR-146a-5p suppressed PDAC cell proliferation and sensitized PDAC cells to gemcitabine chemotherapy [98]. In addition, inhibition of NF-kB by Bufalin, a digoxin-like component of traditional Chinese medicine, can reverse MDR1/P-gp-mediated multi-drug resistance in colorectal carcinoma [99].

### 5.2. FOXO Family of Transcription Factors

The transcription factors of the forkhead box, class O (FOXO) family is also involved in the regulation of the expression of genes coding for antioxidant enzymes, including catalase, sestrins, and SOD1 and 2 [100,101]. FOXO activity can be modulated at multiple levels, including posttranslational modifications, interaction with coregulators, alterations in FOXO subcellular localization, protein synthesis, and stability. Moreover, transcriptional and posttranscriptional control of the expression of genes coding for FOXOs is sensitive to ROS [102]. The transcription factor FOXO3, a member of the FOXO family, is involved in apoptosis induction and drug resistance, and longevity [103]. Although FOXO3 has been characterized as a tumor suppressor gene based on its anti-proliferative and pro-apoptotic functions, an increasing number of studies describe its involvement in chemoresistance of different cancer types, including myeloid leukemia, glioblastoma, pancreatic cancer, and neuroblastoma [104,105,106,107,108,109]. The proposed mechanisms, adopted by FOXO3 to promote chemoresistance, are diverse, and rarely regard the reduction of oxidative stress by FOXO upregulation. In glioblastoma, FOXO3 triggers chemoresistance via the regulation of β-catenin [107]. In glioma stem cells, FOXO3 is involved in radiotherapy resistance, as its inhibition enhances the response to radiotherapy [110]. In metastatic colorectal cancer cells, Yu et al. reported that FOXO3a directly binds to the c-Myc promoter and activated the transcription of the c-Myc gene and thus participated in regulating of c-Myc downstream genes and activating cetuximab resistance [111]. Drug resistance is also mediated by the FOXO–FOXM1 axis, thereby promoting tumorigenesis and cancer progression [112].

### 5.3. AP-1 Transcription Factors

AP-1 represents a family of dimeric transcription factors that exert antioxidant effects through the induction of genes that scavenge ROS, synthesize GSH, suppress levels of free iron, and metabolize pro-oxidant xenobiotics [113]. AP-1 transcriptional activity, which can be activated by H_2_O_2_, cytokines, etc., is mediated mainly by JNK and p38 MAP kinase cascades [114] and regulates a wide range of cellular processes, including proliferation, differentiation, and apoptosis [115,116]. However, the role of AP-1 in tumor development and/or chemoresistance is not clear and depends on the cell type and its differentiation state, tumor stage, and the genetic background of the tumor [116]. Indeed, it has been demonstrated that downregulation of AP-1 expression induced apoptosis and inhibited MMP-2 expression in SCC9 cells in vitro [117]. Moreover, the inhibition of the Raf-1/AP-1 pathway caused a reduction of P-glycoprotein (P-gp) expression and improved the drug sensitivity [118]. The rapid and sustained downregulation of c-fos and the translocation inhibition of its homodimers AP-1 may contribute to the drug-sensitive effects by activating X-linked inhibitor of apoptosis protein (XIAP) in MCF7 breast cancer cells [119]. Although AP-1 can be activated by oxidative stress and exerts antioxidant effects, its role in regulating antioxidant responses in chemoresistance has not yet been fully elucidated.

### 5.4. P53 Transcription Factor

A very important role in maintaining antioxidant levels in the cells is played by mutant p53, which can act by activating Nrf2 or other transcription factors [120]. Recently, Lisek et al. demonstrated that missense mutant protein 53 interacts with Nrf2 and contributes to activating or repressing specific components selectively of its transcriptional program, thereby promoting a pro-survival oxidative stress response that allows cancer cells to survive with high levels of intracellular ROS [121]. The activation of Nrf2 by mutated p53 is associated with poor prognosis in breast cancer patients and includes genes, such as thioredoxin (TRX); in contrast, mutated p53 represses other Nrf2 targets, including heme oxygenase 1 (HMOX1), that can have cytotoxic effects in cancer cells [122]. Mutated p53 promotes cancer cell survival under tumor- and therapy-associated stress conditions by inhibiting the apoptotic and autophagic responses. Moreover, mutated p53 is involved in the resistance to chemotherapy in head and neck squamous cell carcinoma (HNSC) through the inhibition of the p73 pro-apoptotic transcriptional program [123], or the blocking of caspase 9 activity [124] and caspase 9-mediated activation of mitochondrial caspase 3 [125].

### 5.5. PGC-1a and YAP Transcriptional Coactivators

PGC-1a transcriptional coactivator is a master regulator of mitochondrial biogenesis [126]. PGC-1α increases oxidative phosphorylation (OXPHOS) by activating PPARγ and nuclear respiratory factors 1 and 2, and also increases antioxidant capacity and decreases mitochondrial production of ROS by activating uncoupling protein-1 and -2 and stimulating mitochondrial biogenesis [127]. Although the exact role of PGC-1α in cancer is not entirely understood, since it can be upregulated or downregulated in diverse cancer types [128], its role in chemoresistance has been reported. Paku et al. demonstrated that sirtuin 3 (SIRT3) increased colorectal cancer cell chemoresistance through SOD2 and PGC-1α up-regulation, and SIRT3 inhibition sensitized cancer cells to the chemotherapy [129]. Cruz-Bermúdez et al. reported that SCLC cells, with increased PGC-1a, induced by ZLN005 treatment, showed reduced cisplatin-driven apoptosis, whereas PGC-1α interference increased cisplatin sensitivity [130].

Another transcriptional coactivator, yes-associated protein 1(YAP), a key component of the Hippo tumor-suppressor pathway [131], is involved in chemoresistance and in the regulation of oxidative stress. Indeed, we demonstrated crosstalk between Nrf2 and YAP in bladder cancer cells and their involvement in chemoresistance [132]. We demonstrated that the impairing of YAP protein expression reduced GSH content and Nrf2 expression and the silencing of Nrf2, in addition to the depletion of GSH by buthionine sulfoximine (BSO) treatment, inhibited YAP expression. Finally, the silencing of either YAP or Nrf2 enhanced the sensitivity of chemoresistant bladder cancer cells to cytotoxic agents. In addition, YAP can activate the FOXO1 transcription factor to form a functional complex on the promoters of antioxidant genes and, through this mechanism, increase the antioxidant capacity of the cell. As a consequence, the inactivation of YAP, induced by Hippo activation, suppresses FOXO1 activity and decreases antioxidant gene expression [133]. Other authors reported that YAP silencing significantly enhanced autophagic flux by increasing RAC1-driven ROS and reduced the resistance to chemotherapeutic agents in hepatocarcinoma cells [134]. The involvement in chemoresistance of YAP has also been demonstrated by Bauzone et al., who found a crosstalk between YAP and retinoic acid (RAR)–retinoid X receptor (RXR), which promoted 5-fluorouracil resistance and self-renewal in colorectal cancer cells [135]. Analogously, the interaction between phosphatase SHP2 and YAP was involved in the chemoresistance of cholangiocarcinoma cell lines [136]. Moreover, Santoro et al. found that the YAP expression was suppressed, and the malignancy and treatment resistance of pancreatic cancer cells were reduced, after inhibition of glycogen synthase kinase (GSK3) [137].

## 6. Pro-Oxidant Drugs and Antioxidant-Dependent Chemoresistance

Several pro-oxidant chemotherapeutic drugs can induce acquired resistance by stimulating antioxidant responses in cancer cells. Among these prooxidant drugs, an important place is occupied by doxorubicin. Doxorubicin, widely used to treat various cancers such as brain tumors, thoracic cancers, breast, and esophageal carcinomas, etc., principally acts by inhibiting DNA topoisomerase II activity by forming DNA adduct and enhancing the production of ROS, which are important in triggering apoptosis through the mitochondrial pathway [138]. Recently, it has been demonstrated that doxorubicin chemoresistance also depends on its ability to stimulate Nrf2 signaling, thus protecting cells against cell death [139].

Cisplatin (CDDP) is one of the most widely used platinum-based compounds for the treatment of various cancers, including, ovarian cancer, bladder cancer, head and neck cancer, and NSCLC. Despite a consistent rate of initial responses, cisplatin treatment often results in the development of chemoresistance. The cisplatin cytotoxic effect predominantly includes DNA damage, and, as a consequence, a blocking of cell division and an induction of apoptosis. All these events are accompanied by an increase in oxidative stress [140] due to cisplatin interaction with endogenous nucleophiles, including GSH, methionine, metallothionein, and other cysteine-rich proteins, resulting in the depletion of antioxidant reserves and the increase of oxidative stress [141,142]. In this context, the inactivation of cisplatin action also depends on the acquired high expression of antioxidant protein networks, which promote drug resistance [143,144].

Arsenic trioxide (As_2_O_3_) is an anti-cancer drug approved by the US Food and Drug Administration to specifically treat acute promyelocytic leukemia, and it is also used to treat other malignancies, such as multiple myeloma and myelodysplastic syndromes [145]. Several types of evidence indicated that As_2_O_3_ causes oxidative stress and, through this stimulus, can affect signal transduction and activation of transcription factors, associated with the redox condition of the cells, resulting in the stimulation of apoptosis [146]. In NSCLC, the resistance to As_2_O_3_ therapy seems to be associated with enhanced Nrf2 activity [147,148]. Moreover, it has been demonstrated that the cytotoxic effect of As_2_O_3_ was decreased in acute promyelocytic leukemia cells treated with carnosic acid, a typical activator of Nrf2 [149].

Paclitaxel is a microtubule-stabilizing chemotherapeutic drug. Its antitumor mechanisms are related to the disassembly of microtubules [150] and the release of ROS from mitochondria [151]. It is used for the treatment of several cancers, including breast cancer, gastric cancer, NSCLCs, and cervical carcinoma. Paclitaxel resistance frequently occurs after paclitaxel treatment. After paclitaxel chemotherapy, patients showed a higher expression of antioxidant enzymes, such as HO-1 and serine hydroxy-methyltransferase, resulting in a decrease in ROS levels, which leads to chemoresistance [152].

Gemcitabine (2′,2′-difluoro 2′-deoxycytidine), a deoxycytidine analog, is widely used in the treatment of various solid tumors, in particular pancreatic cancer [153]. However, pancreatic cancer cells, which at first were sensitive to gemcitabine, developed resistance within a few weeks after treatment [154]. It has been widely demonstrated that the increase of antioxidants is closely involved in the chemoresistance of gemcitabine. Zarei et al. demonstrated that the overexpression of the antioxidant enzyme dehydrogenase 1 (IDH1) was sufficient to fully restore gemcitabine resistance in pancreatic ductal adenocarcinoma (PDAC) by reducing ROS levels [155]. Zhao et al. demonstrated that low ROS levels maintain the stemness and epithelial–mesenchymal transition (EMT) phenotypes in gemcitabine-resistant PADC cells [156]. Analogously with that observed in doxorubicin-treated cancer cells, the gemcitabine resistance also depends on an increase in antioxidant genes Nrf2, SOD1, SOD2, catalase, and glutathione peroxidase 1 [157].

Bortezomib (BTZ) is a proteasome inhibitor used as a single agent or in combination with conventional drugs, such as dexamethasone, in the treatment of multiple myeloma patients [158,159]. Proteasome inhibition results in the accumulation of unfolded proteins, which trigger endoplasmic reticulum (ER) stress, which has been shown to cause cell death via multiple pathways including overproduction of ROS [160,161]. In bortezomib-resistant myeloma cells, HO-1 mRNA levels were increased, compared to parent cells, and HO-1 inhibition restored the sensitivity to bortezomib [162]. Tibullo et al. showed that the increase of ROS formation by bortezomib treatment was accompanied by a significant overexpression of HO-1, and the inhibition of HO-1 nuclear translocation increased sensitivity to bortezomib [163].

## 7. Nrf2 Inhibitors in the Treatment of Chemoresistant Tumors

The increase of antioxidant defenses observed in several chemoresistant cancers led to the development of diverse research lines aimed to identify new compounds for treating cancers that do not respond to conventional therapy. In particular, some inhibitors of Nrf2 and its target products have been investigated [164]. Historically, some natural compounds have been used to potentiate the efficacy of ROS-inducing agents against chemoresistant cancers. One of the first natural compounds studied was brusatrol, a quassinoid from the plant *Brucea javanica* [165]. Brusatrol inhibits the Nrf2 expression and transcriptional activity and overcomes resistance to several chemotherapeutics of tumors and cancer cell lines [166].

Ailanthone is a natural compound extracted from the tree *Ailanthus altissima*. Many reports demonstrated that Ailanthone was able to inhibit the proliferation of different cancer cell lines such as melanoma [167], acute myeloid leukemia [168], gastric [169], lung [170], and breast cancer [171]. Ailanhone can reverse cancer resistance by affecting different pathways [172,173], including the inhibition of Nrf2 and YAP pathways [42,43].

Halofuginone, an Nrf2 inhibitor from the plant *Dichroa febrifuga*, has been shown to reverse the radioresistance of Lewis lung cancer cells [174] and enhance the chemo-sensitivity of cancer cells both in vitro and in vivo [175].

In recent years, some other molecules able to inhibit Nrf2 have been identified [176]. From a screening of a chemical library of 400,000 molecules, the main compound found able to block Nrf2 was ML385. ML385, by interacting with ARE sequences, prevents the binding of Nrf2 to AREs, inhibits Nrf2 transcriptional activity, and overcomes the resistance of Keap1-deficient NSCLC cells to carboplatin and other chemotherapeutics [176]. Another Nrf2 inhibitor is AEM1, which, by inhibiting Nrf2 transcriptional activity, sensitized Keap1-deficient A549 lung tumor cells to various chemotherapeutic drugs [177].

Other Nrf2 inhibitors, such as all-trans retinoic acid (ATRA), clobetasol propionate (CP), and apigenin are under clinical investigation. ATRA markedly reduces the ability of Nrf2 to mediate induction of ARE-driven genes, thus blocking the activation of the Nrf2 pathway [178].

CP prevents nuclear accumulation and promotes the degradation of Nrf2 in a glucocorticoid receptor and a GSK-3 dependent way, thus potentiating the therapeutic efficacy in Keap1 mutant lung cancer [179]. CP was introduced in a phase 2 clinical trial (NCT02368886), together with regorafenib, in treating patients with refractory metastatic colorectal cancer.

Apigenin sensitizes doxorubicin-resistant hepatocellular carcinoma cells to doxorubicin through the inhibition of the PI3K/Akt/Nrf2 pathway [180] and sensitizes pancreatic cancer cells, to chemotherapy [181]. However, in addition to Nrf2 inhibition, apigenin affects molecular pathways such as those associated with the hypoxia-inducible factor (HIF), vascular endothelial growth factor (VEGF), and glucose transporter-1 (GLUT-1) [182,183]. Apigenin reached a clinical trial in 2008 (NCT00609310) in combination with epigallocatechin gallate in the treatment of disease relapse in patients with colorectal cancer, investigating the prevention of neoplasia recurrence. However, this study has been suspended. Overall, it was found that bioactive compounds can affect different metabolic pathways to exert their toxic action in tumor cells, thus causing some side effects. For this reason, further studies are needed to insert Nrf2 inhibitors in clinical trials.

## 8. Conclusions

In recent years, studies concerning the role of oxidative stress control in the chemo- and radioresistance of different types of tumors and the identification of proteins involved in these processes as possible pharmacological targets have increased enormously. In this context, the Nrf2 pathway has gained great attention, and in vitro studies have shown that its inhibition can sensitize several types of cancer cells to chemotherapy or radiotherapy. However, in vivo studies of this therapeutic approach are still few, and will require further efforts in the coming years.

Certainly, a lot of ground has been covered since the first studies on oxidative stress and lipid peroxidation were carried out by pioneers such as Professors Mario U. Dianzani, Trevor F. Slater, and Richard O. Recknagel, but their insights have allowed the scientific community to advance in a number of research fields and to develop new therapeutic strategies, which will allow, in the coming years, advancing the possibility of treating tumors more resistant to conventional therapy.

## Figures and Tables

**Figure 1 antioxidants-10-00510-f001:**
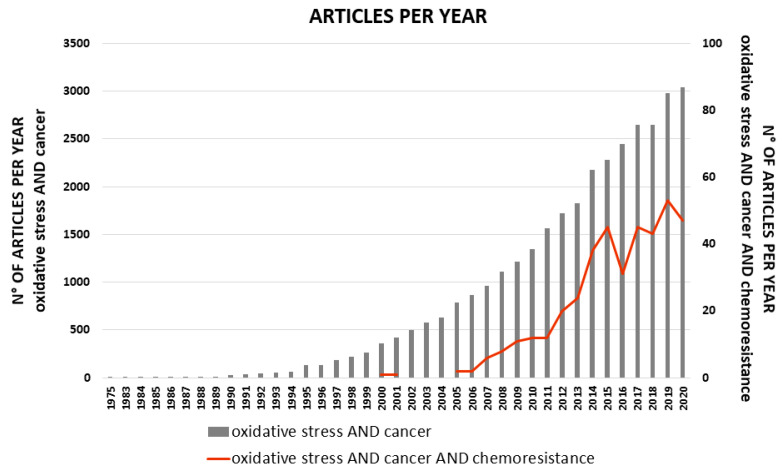
Number of articles per year, reported on PubMed, entering the search filter: “oxidative stress AND cancer” (bar graph; *Y*-axis on the left side) or “oxidative stress AND cancer AND chemoresistance” (line graph; *Y*-axis on the right side) in title and/or abstract.

**Figure 2 antioxidants-10-00510-f002:**
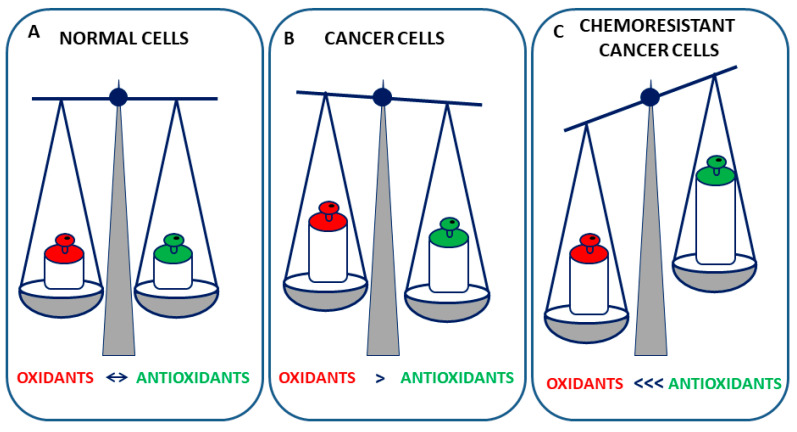
Oxidant and antioxidants in redox homeostasis in normal, cancer, and chemoresistant cancer cells. (**A**) Normal cells keep constant oxidant production and elimination to maintain a favorable redox balance. (**B**) Cancer cells exhibit higher steady-state levels of oxidant species, counterbalanced by increased antioxidant capacity. (**C**) Chemoresistant cancer cells up-regulate the antioxidant defenses to counteract oxidant species.

**Figure 3 antioxidants-10-00510-f003:**
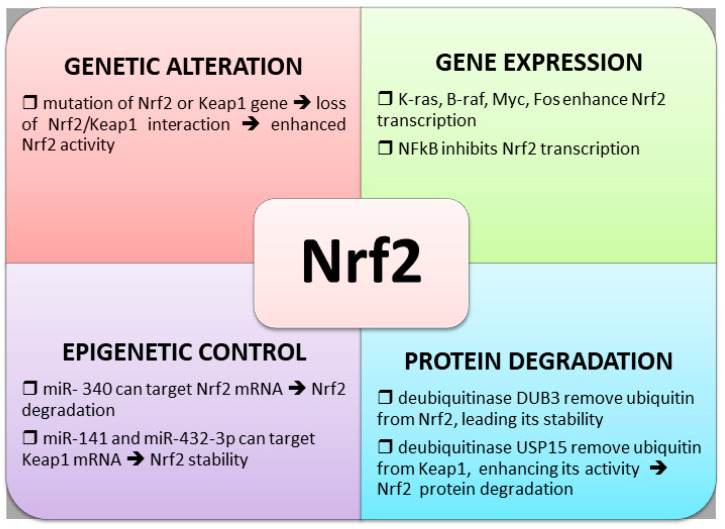
Mechanisms of control of Nrf2 activity.

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
