# Peer review of "Control of Oxidative Stress in Cancer Chemoresistance: Spotlight on Nrf2 Role"

_antioxidants, 2021, doi:10.3390/antiox10040510_

Round 1

Reviewer 1 Report

Barrera et al. present in this manuscript an accurate and adequate overview of the present knowledge about the possible role(s) of redox equilibria and oxidative processes in the chemoresistance of cancer cells. The paper is clearly and concisely written, with an updated list of references. As such the paper deserves consideration. Minor: the use of colour in the illustrations (simples cartoons) appears not to be sufficiently warranted, the use of a plain b&w style would be probably preferable.

Author Response

We thank the Reviewer 1 for accurate and helpful revision of our manuscript. Please, find enclosed the answers to your question.

Barrera et al. present in this manuscript an accurate and adequate overview of the present knowledge about the possible role(s) of redox equilibria and oxidative processes in the chemoresistance of cancer cells. The paper is clearly and concisely written, with an updated list of references. As such the paper deserves consideration. Minor: the use of colour in the illustrations (simples cartoons) appears not to be sufficiently warranted, the use of a plain b&w style would be probably preferable.

The colors of the illustrations have been changed. In fig.1 and 2, blue has been replaced with gray and in fig.,3 the colors too bright have been replaced with softer colors. I hope that the colors of the illustrations are now warranted.

Reviewer 2 Report

In this review by Barrera et al., the authors have provided a collective information on the role of oxidative stress in chemoresistance of cancer cells. This is an interesting topic. The manuscript can be reconsidered for publication after addressing the below points,

1. In figure 1, there is a typo error on the legend of y-axis.

2. The authors should provide more details on the chemotherapy induced resistance in relation to oxidative stress.

3. An additinonal topic on the drugs that target oxidative stress pathways and clinical trials would improve the manuscript.

4. The authors focused more on the nrf2 pathway and chemoresistance.  So, the title of the manuscript can be changed accordingly.

5. ROS induced autophagy and chemoresistance should be included.

Author Response

We thank the Reviewer for accurate and helpful revision of our manuscript. The manuscript has been extensively revised and new paragraphs have been added to improve the review. The new parts are written in red.
Please, find enclosed point by point the answers to your questions.

  1. In figure 1, there is a typo error on the legend of y-axis.

Fig. 1 has been corrected.

  1. The authors should provide more details on the chemotherapy induced resistance in relation to oxidative stress.

A paragraph regarding the acquired resistance after cancer treatment with pro-oxidant chemotherapeutic drugs has been added.

  1. An additional topic on the drugs that target oxidative stress pathways and clinical trials would improve the manuscript.

An additional paragraph on the drugs that target oxidative stress pathways and clinical trials has been added.

  1. The authors focused more on the nrf2 pathway and chemoresistance.  So, the title of the manuscript can be changed accordingly.

The title has been changed. Now the new title is: “ Control of oxidative stress in cancer chemoresistance: spotlight on Nrf2 role”.

  1. ROS induced autophagy and chemoresistance should be included.

In the paragraph: “Oxidative stress in cancer cells” we have inserted some data about the role of autophagy as a mechanism of cell death-induction by ROS.

Reviewer 3 Report

The presented to review manuscript concerns the critical element hindering effective of cancer treatments - chemoresistance. The paper is presented in an interesting and way. But, it focuses mainly on the Nrf2 transcription factor. This is a significant contraindication to accepting the work in the present form.
I suggest that the modification of the manuscript title should according to the discussed subject. If the authors maintain the willingness to present the role of other factors, they should be described in more detail. Especially in the chapters about FOXO and AP-1, the given knowledge is very superficial and not cohesive with the article title. I  strongly suggest the authors rewrite the manuscript according to the above suggestions to achieve the intended goal of the paper, and also because the review paper should refer to the complete knowledge of the analyzed topic.

Author Response

We thank the Reviewer for accurate and helpful revision of our manuscript. The manuscript has been extensively revised and new paragraphs have been added to improve the review. The new parts are written in red.
Please, find enclosed the answer to your questions.

The presented to review manuscript concerns the critical element hindering effective of cancer treatments - chemoresistance. The paper is presented in an interesting and way. But, it focuses mainly on the Nrf2 transcription factor. This is a significant contraindication to accepting the work in the present form.
I suggest that the modification of the manuscript title should according to the discussed subject. If the authors maintain the willingness to present the role of other factors, they should be described in more detail. Especially in the chapters about FOXO and AP-1, the given knowledge is very superficial and not cohesive with the article title. I  strongly suggest the authors rewrite the manuscript according to the above suggestions to achieve the intended goal of the paper, and also because the review paper should refer to the complete knowledge of the analyzed topic.

We recognize that our review is primarily focused on the role of Nrf2 in chemoresistance. For this reason the title has been changed. Now the new title is: “ Control of oxidative stress in cancer chemoresistance: spotlight on Nrf2 role”.

We admit that the space dedicated to other transcription factors, such as AP1 and FOXO, is rather limited. They have been cited because they can be regulated, or are involved in the regulation of oxidative stress, but most of the data concerning their role in chemoresistance do not include the regulation of oxidative stress, but other pathways, which are equally important, but beyond the scope of the review.

We hope that the new title of the revised review justifies this choice.

Round 2

Reviewer 2 Report

The manuscript was improved substantially by revision and can be considered for publication after addressing the below point.

1. The legend of figure 1 Y axis, "No of articles per year..." should be changed to "Number of articles per year..."

Reviewer 3 Report

As the authors have specified the manuscript's subject, it is now in line with the title. The additional information contained in the paper enriched it and made it even more complete. Therefore, I strongly recommend the revised version of the manuscript for publication in Antioxidants in its present form.